# Noninvasive Subcellular Imaging Using Atomic Force Acoustic Microscopy (AFAM)

**DOI:** 10.3390/cells8040314

**Published:** 2019-04-05

**Authors:** Xiaoqing Li, Ang Lu, Wenjie Deng, Li Su, Jing Wang, Mingyue Ding

**Affiliations:** 1Department of Biomedical Engineering, College of Life Science and Technology, Huazhong University of Science and Technology, Wuhan 430074, China; d201677426@hust.edu.cn (X.L.); wenjie.deng@dental360.cn (W.D.); 2Department of Biophysics, College of Life Science and Technology, Huazhong University of Science and Technology, Wuhan 430074, China; luang199004@163.com (A.L.); lisu@hust.edu.cn (L.S.)

**Keywords:** atomic force acoustic microscopy, image fusion algorithm, parameters optimization, subcellular structural image atlas

## Abstract

We report an imaging approach applying the atomic force acoustic microscopy (AFAM), which has unique potential for nondestructive imaging of cell internal structures. To obtain high spatial resolution images, we optimized the significant imaging parameters, including scanning speeds, feedback configurations and acoustic frequencies of an AFAM system, to increase the amplitude of the acoustic signal and to stabilize the morphological signals. We also combined the acoustic amplitude and phase signals, and generated pseudo-color figures for better illustration of subcellular features such as pseudopodia, membranes and nucleus-like. The subcellular structural image atlas can describe nanoscale details of multiple samples and provide clearer images of the subcellular features compared to other conventional techniques. This study builds a strong basis of transmission AFAM for cell imaging, which can help researchers to clarify the cell structures in diverse biological fields and push the understanding of biology evolution to a new stage.

## 1. Introduction

A comprehensive characterization of subcellular structures remains a challenge for biological research. The conventional optical imaging methods have advanced to achieve cell images at nano-resolution, but are limited to some complexities, such as the fluorophore localization and fitting algorithms to correct for sample drift during acquisition [1]. Scanning probe microscopy has advanced the nanoscale imaging for bio-samples due to its strong capabilities of mapping the morphological and mechanical features of subcellular structures with extreme high spatial resolution [2]. In particular, atomic force microscopy (AFM) can acquire topographical surface images as well as the mechanical properties of biological samples at nanoscale resolution by measuring the forces acting between surface chemical groups [3,4,5,6]. Various improvements have been made to the AFM platform to acquire more quantitative, dynamic and chemical information, which can be accurately extracted from the acquired images [7,8,9,10,11,12]. Another powerful approach for nanoscale imaging, scanning electron microscope (SEM), provides methods to characterize the inner architecture of a single soft material nanoparticle in addition to its size and shape [13]. Transmission electron microscopy can be used to visualize some virus particles and cells at nano-resolution [14]. Soft X-ray tomography can also be used for the quantitative imaging of organelle structure and distribution in whole cells [15]. Although those techniques have achieved nanometer resolution on cell imaging, they both have their own limitations. For instance, AFM images have insoluble artifacts caused by the abrupt height of the samples and can only obtain the surface or sub-surface structure [7,8,9,10,11,12]; SEM and TEM have special requirements for sample preparation [13,14]; and soft X-ray tomography has potential radiation safety limitations [15].

Among all biomedical imaging techniques, ultrasound has played a significant role in the assessment, diagnosis and imaging of organs and tissues due to its noninvasive natures. The use of ultrasonic waves to explore the inner structures of materials emerged with the introduction of scanning near-field ultrasound holography (SNFUH) [16]. This technique utilized the acoustic resonance to provide lateral as well as depth-based imaging of nanostructures including alveolar macrophages and peripheral erythrocytes [17,18,19,20]. However, fast energy attenuation of standing waves limited the application of ultrasound waves to image thin samples tens of microns thick.

Recently, AFAM was developed by combining ultrasound imaging with the near-field AFM imaging. In contrast to SNFUH, in the transmission mode of AFAM, the single frequency acoustic wave penetrates through the specimens and is detected by the high-frequency vibration of the probe. Hence, the transmission mode of AFAM allows deeper imaging while still maintaining nanoscale resolution. In addition, atomic force acoustic microscopy can not only characterize the surface information of various cells at the submicron scale but also image interior substructures by measuring changes in the phase and amplitudes of acoustic waves. The acoustic wave phase represents the position of a wave at a particular time in its cycle, and the relative phase difference indicates that there are certain substances in the path to accelerate or decelerate the propagation of sound waves. The acoustic wave amplitude reflects the wave attenuation along the propagating direction due to the energy dissipation. Hence, the changes of amplitude and phase can explore the internal structure information.

In this paper, we present a novel method to acquire cell images with nanoscale resolution in a non-invasive and easily handled manner (Figure 1). According to the multiple repeated experiments, we optimized the scanning speed, feedback configurations and acoustic frequency of the AFAM to strengthen the acoustic signals and to stabilize the morphological signals. We also established an RGB-fusion model to illustrate the subcellular features by combining different information from the original images. Applying the optimized AFAM setup and an RGB-fusion algorithm, we obtained submicron-scale images of various types of cells, including *Escherichia coli* (*E. coli*), *Staphylococcus aureus* (*S. aureus*), onion epidermis, MCF7 breast cancer cells, MDA-MB-231 breast cancer cells and human erythrocytes, which proved the adaption capacity and robustness of this technique. These experiments revealed distinct morphology and indicated the internal substructures of both eukaryotes and prokaryotes. The fused images enabled better visual observations of morphology and subcellular structures.

## 2. Materials and Methods

AFAM imaging: CSPM5500, an open atomic force acoustic microscope, from Being Nano-Instruments Ltd., Guangzhou, China, was used as the measurement platform. The ultrasound transducer under the sample holder/plate was used to launch a single-frequency ultrasound wave and was coupled with a normal ultrasound coupling agent as a glue-like substance, to make the specimen contact the sample holder/plate closely. A scanning acoustic model, known as transmission mode, was applied, in which the acoustic wave penetrated through the samples. A contact probe, obtained from Budget Sensors, with a resonant frequency of 13 kHz and a force constant of 0.2 N/m, was used for imaging. The probe tip scanned across the surface of cells on a silicon (Si) wafer, while the acoustic images were acquired with the morphological images simultaneously. The low-frequency vibration represented the morphological information, and the relatively high-frequency vibration showed signals of the acoustic images. All the experiments were performed at room temperature (22 °C).

All parameter-optimization experiments used the MDA-MB-231 cells. We first tested the imaging speed effects on MDA-MB-231 cell imaging, varying the scanning frequency from 0.1 to 2 Hz at 1024 data points per scanning line. Then, we changed the feedback configurations of the integral gain (100–300 times), proportional gain (100–300 times) and reference point value (0.10–0.20 V) to choose the most suitable feedback configurations. We selected 1.5 MHz and 30.5 kHz, which produced the best acoustic response, to illustrate the influence of the acoustic frequency.

Samples preparations for atomic force acoustic microscopy measurements: MDA-MB-231 and MCF7 cells, two classic types of breast cancer cells that play important roles in exploring the biochemical mechanisms of breast cancer, stored by the China Center for Type Culture Collection in Wuhan (Wuhan H22), were grown on a Si wafer in Dulbecco’s modified Eagle’s medium (DMEM) supplemented with 10% fetal bovine serum (FBS), 50 U/mL penicillin and 50 U/mL streptomycin at 37 °C in 95% humidified air with 5% CO_2_. Before AFAM measurement, the cells on the Si wafer were treated with a 4% solution of paraformaldehyde for 15 min to fix the cells, which can maintain the cell morphology, make the cells adhere to the wafers more firmly to prevent them from being taken away by the probe during the imaging scan and further make the scanning process less affected by the sample shift. The fixation treatment with 4% solution of paraformaldehyde has been widely used in other optical imaging approaches, e.g. fluorescence imaging. There are no significant artifacts created in the procedure, which has been reported [21]. After fixation, the cells were washed with ultrapure water to eliminate the salt crystal and air-dried naturally. Onion epidermis cells, widely used to observe the structures of plant cells under optical microscopes, were placed on a Si wafer after tearing off a small piece from the fresh root top of an onion. The morphology, structure and function of erythrocytes, or red blood cells, are active research areas because these cells carry oxygen to satisfy the respiration of various parts of the body. After resting at 4 °C for 4 h, fresh normal human erythrocytes extracted from human whole blood, which was generously provided by the volunteer in our laboratory, were spread on a Si wafer. Onion epidermis cells and human erythrocytes were applied to a Si wafer using micro pipette tips and dried naturally prior to AFAM measurement. *Escherichia coli* (*E. coli*) and *Staphylococcus aureus* (*S. aureus*) are two main species of contaminating bacteria that cause inflammatory reaction. *E. coli* and *S. aureus*, stored by the China Center for Type Culture Collection in Wuhan (Wuhan H22), were taken out of the −20 °C refrigerator. After cultured in Luria–Bertani (LB) media in an incubator shaker at 37 °C and 250 rpm for 12 h, the bacterial cells were in the log phase. A total of 50 μL medium of bacteria was added to a 1.5 mL EP tube and centrifuged for 4 min at 10,000 rpm/min and 4 °C. One milliliter of ultrapure water was added, and the solution was blended. The bacteria medium was applied to a Si wafer and air-dried naturally, and the Si wafers were exposed to an open flame for 0.2 s prior to AFAM measurement. We chose the six samples mentioned above to test the imaging capacity of our method, in which MDA-MB-231, MCF7 cells, *E. coli* and *S. aureus* were fixed, while the onion epidermis cells and erythrocytes were unfixed and native. The treatment with 4% solution of paraformaldehyde or the flame preprocessing aimed to fix the cells and keep their morphology during the imaging scan, which is a preprocessing of cells cultured in medium. In other words, this imaging approach does not require the cell fixation as a necessary process step, which we checked though a supplemented experiment. We fixed the cells to follow the common cell preparation, which kept the cell shape so that it can be easily compared with other imaging modality.

Samples preparations for laser scanning confocal microscope and scanning electron microscope measurements: Onion epidermis cells were placed on a microslide and photographed by an inverted fluorescence microscope (NIKON ECLIPSE Ti-S). MDA-MB-231 cells were first fixed by 4% solution of paraformaldehyde for 15 min, then stained by 4′,6-diamidino-2-phenylindole (DIPA) for 3 min, washed by phosphate buffer saline (PBS), and finally photographed by laser scanning confocal microscope (OLYMPUS FV1000). MCF7 cells were photographed by laser scanning confocal microscope (OLYMPUS FV1000) after fixation by 4% solution of paraformaldehyde for 15 min. Fresh normal human erythrocytes were first diluted 10 times, then applied to a microslide and photographed by laser scanning confocal microscope (OLYMPUS FV1000). We strictly followed the procedures described in Ref. [18], to get the SEM images of *E. coli* and *S. aureus* [22].

Data analysis: The original AFAM images were visualized with the software Imager in the AFAM system, which is called CSPM5500. Usually, each individual image of AFAM can only provide one restricted and specified view of the surface or the information of internal structures. In traditional AFAM experiments, the morphological and acoustic images are displayed separately, and it is difficult to distinguish structural information differences and overlaps among these obtained images. One possible solution is to integrate data from multiple images, creating a single fused image that retains all the features of the original sources. RGB (red, green and blue) color images are a special case of multispectral fused images that utilize the three basic colors of human vision. An image fusion algorithm based on RGB colors could be used to show the complementary information of all source images and provide a stereoscopic and colorful fused figure. The fusion of the RGB images comprised three steps. A normalization process was performed on three source images—the morphological image, phase image and amplitude image—to obtain gray images at the pixel level. A normalization process was performed using Equation (1),
(1)j=(i/max)∗256
where *i* is the value of every pixel and max is the largest value of all pixels. Histogram modification was performed to maximize the image contrast using Equation (2),
(2)$$S=T(rk)=\sum_{i=0}^{k}(nk/N)=\sum_{i=0}^{k}pr(ri)$$
where *N* is the total number of pixels, *n_k_* is the value of the pixel at the *k*th level, *r_k_* is the grayscale at kth level, and *p_r_* (*r_i_*) is the relative frequency at the *i* level. To apply the RGB model, the morphological image was taken as the red channel, the phase image of the ultrasound as the green channel, and the amplitude image of the ultrasound as the blue image of the RGB color system to integrate the three images into one color image.

## 3. Results

### 3.1. Transmission Mode of Atomic Force Acoustic Microscope System

To obtain the internal structure information of samples and their morphological surface characteristics simultaneously, we combined an ultrasound transducer to a commercial AFM platform and built an atomic force acoustic microscopy system, as shown in Figure 1. Similar to AFM, the surface morphology of samples were acquired via the displacement of a cantilever with a sharp probe tip at its end. As the tip scanned along the specimen surface, a laser beam from the laser diode module was reflected by the cantilever oscillating accordingly and caught by the position-sensitive detector. This AFM-like system operated in constant force mode, where the piezoelectric scanner manipulated the sample holder up and down to maintain the laser spot at the center of a position-sensitive detector. During AFM morphological measurements, an ultrasound transducer attached to the piezoelectric scanner emitted single-frequency acoustic waves that travelled vertically upwards through the sample. This transmitted ultrasound then irradiated the probe tip with high-frequency vibrations, which could be recorded to recover the changes in the amplitude and phase of the longitudinal acoustic wave. Two-dimensional distributions of acoustic amplitudes and phases mapped the delicate surface and internal structures. The silicon tip had a diameter of approximately 10 nm at the pinpoint and could therefore ideally detect surface changes at a 10 nm scale. In our preliminary work, we imaged 20 nm gold nanoparticles buried in a 500 µm polymer cover layer. The particles and their three-dimensional distribution are clearly visible in our acoustic images [23].

### 3.2. Parameters-Optimization of AFAM System

The resolution of AFAM imaging is determined by the scanning speed of the probe across the surface, the amplification of laser signals, and the sensitivity of the probe material to acoustic frequencies. To improve the surface and internal topological image qualities, we optimized the key parameters of the AFAM system, including the scanning speed, feedback configurations and acoustic frequency. In Figure 2 and Appendix A, images of MDA-MB-231 cells are shown to demonstrate the effects of modifying these parameters in the AFAM system. Through the multiple repeated experimental iterative optimization, it was concluded that the 0.5 Hz scanning frequency produced morphological images of the MDA-MB-231 cell with sharper edges, clearer details and fewer blurring artifacts (Figure 2a,d and Appendix A). The key feedback configurations to improve the measuring accuracy, including multiple interlinked variables such as the integral gain, proportional gain and laser feedback conditions, were also fine-tuned to produce amplitude images of an MDA-MB-231 cell with less noise and sharper edges than those obtained with the non-optimized parameters (Figure 2b,e and Appendix A). A megahertz (MHz) acoustic wave can provide nano-resolution but fast signal attenuation, whereas a kilohertz (KHz) frequency has a deeper penetration depth and maintains submicron resolution. The phase images of 30.5 KHz frequency ultrasound waves showed clearer contours and less noise (Figure 2c) than those obtained at the 1.5 MHz frequency (Figure 2f). Amplitude images showed the same trend, while the morphological images were unchanged (Appendix A). The experimental data show that optimization of the scanning speed, feedback configurations and acoustic frequency in the AFAM system produced images with clearer edges, fewer artifacts, less noise and more details of morphology and subcellular structures.

### 3.3. Analysis of the Acoustic Phase Image

During initial trials of the optimized system, the phase images contained many tiny dot-like speckles that were invisible in the morphological results (Figure 2c). To determine whether these dots were real features or merely noise, we performed morphological and acoustic phase imaging of *Escherichia coli* (*E. coli*), as shown in Figure 3a,d. Enhancing the morphological image contrast by 100 times enabled the observation of faint dot-like structures in the background, including a very bright spot and several arranged along a nearly straight line (Figure 3b). The corresponding region in the phase image also contains these faint structures, while the size, quantity and arrangement were consistent with those in the morphology images (Figure 3e). In the *E. coli* body structures, the phase image not only showed body boundary, but also gave more fine details of many dots with different phase shifts compared to the morphological image, which suggested some internal structures. For example, the upper *E. coli* body showed a small thin region in the middle of the morphological image, indicated by an arrow (Figure 3c); in the same position, the phase image had a large and characteristic fuzzy area (Figure 3f), which might suggest some intracellular macromolecule changes during the division period. This result demonstrates that phase images could enhance the capability of distinguishing faint details and indicate subcellular structures.

### 3.4. Image Fusion Algorithm

For better characterization of the morphological and acoustic images, we developed a fusion algorithm based on RGB pseudo-coloring. The framework of the proposed method is depicted in Figure 4. We first normalized each of the three source images to its maximum, obtaining three grayscale images with similar brightness, and then assigned them to the three color channels: the morphology image to red, the amplitude image of the ultrasound to green, and the phase image of the ultrasound to blue. The resulting RGB fused images showed unique signals from each component, while superimposed color indicated common information.

The multi-color fused image of an MDA-MB-231 cell acquired by AFAM (Figure 5a–d) compared favorably to fluorescence images (Figure 5e,f) of the same bio-structures. The fused MDA-MB-231 images showed nuclear regions in the similar positions with relative matched proportions of the cell body (Figure 5d). Indeed, Figure 5d shows a complicated pseudopodium and a clear membrane, whereas fluorescence images (Figure 5e,f) showed only the same structures with rough edges.

### 3.5. Cell Atlas Establishment

A cell atlas collects maps of different types of subcellular features into a database to study their structural functions and provides a basis for biological research. With a parameter-optimized system and multi-colored imaging capabilities, AFAM was applied to establish an atlas of eukaryotes and prokaryotes with nanoscale resolution. Figure 6 shows the AFAM images of six different cell types: onion epidermis cells, MCF7 cells, MDA-MB-231 cells, erythrocytes, *E. coli* and *S. aureus*.

Figure 6 presents an atlas of morphological images, amplitude acoustic images, phase images, fused images and reference images of the six cell types. We obtained typical optical images of plant cell and animal cells (Figure 6Ie–IVe) and SEM images of bacteria (Figure 6Ve,VIe) as references.

The multi-color presented AFAM images showed significant differences between cells with or without walls. For example, the figures of onion epidermis cells and two kinds of bacteria, two classic model cells with walls, showed smooth and clean cell edges (Figure 6I,V)). The onion epidermis cells showed a rectangular-like shape with a cell wall (Figure 6Ia–Ic). The AFAM fused image showed a red cell wall with a certain thickness of 16 μm, compared to the optical image (Figure 6Id,Ie). *E. coli* was a bacillus without spore and blunt circles at both ends with a diameter of approximately 2–4 μm (Figure 6Va–Vc). *S. aureus* has a Gram-positive coccus, with small pathogenicity and a diameter of approximately 0.2–0.5 μm, which was arranged in the form of a globular grape with no spore or capsule (Figure 6 VIa–VIc). Compared to SEM, the AFAM fused image of the two bacteria shown in color and stereoscopic vision showed the typical rod-shaped and circular forms at submicron scale resolution (Figure 6Vd,Ve,VId,VIe)). Erythrocytes and breast cancer cells, on the other hand, supported only by soft cell membrane and cytoskeletons, appeared to have rough edges with varied shapes (Figure 6IV,III). The MCF7 cell had leg-like protrusions known as parapodia approximately 90 μm long, and the phase image suggested complicated internal structures (Figure 6IIa–IIc)). The MDA-MB-231 cell showed a classic fusiform shape, and the phase image showed more details on the cell body, which may contain some subcellular structural information (Figure 6IIIa–IIIc). Compared to optical images, the AFAM fused images of the two breast cancer cells revealed more complicated parapodia and clearer contours, while indicating some internal structures, including ingredient distribution and change of substance (Figure 6IId,IIe,IIId,IIIe). The erythrocytes were disc-shaped and concave in the central section with a thicker edge and a small diameter of approximately 8 μm (Figure 6IVa–IVc). While it was difficult to observe the concave in the central section under a laser scanning confocal microscopy, the AFAM fused image showed a clear classic disc shape (Figure 6IVd,IVe).

The AFAM images also showed significant different acoustic wave behavior in cells with and without a nucleus. For example, erythrocytes, with no cell nucleus, were depressed in the middle (Figure 6IVd); the nucleus of MCF7 cells, however, made them tall in the central section (Figure 6IId). Nucleic acid also impacted AFAM imaging. *E. coli* and *S. aureus* have no cell nucleus but have nucleic acid, while erythrocytes have much less nucleic acid. There were exhibited more complicated phase images and fused figures of these two bacteria (Figure 6Vd,VId) than erythrocytes (Figure 6IVd). Hence, the experimental data above illustrate the strong capabilities to distinguishing certain subcellular structures from the AFAM images.

## 4. Discussion

Technology to non-invasively acquire cell images at nanoscale resolution is essential to meet the demand for characterizing subcellular structures. AFAM can extensively reduce the artifacts and avoid complex sample preparation associated with SEM and AFM. The acoustic images of prokaryotes and eukaryotes acquired by AFAM showed clearer exterior contours than the morphological images compared to the alternative techniques (Figure 6). Moreover, the samples for AFAM measurements are easy to prepare. For instance, *E. coli* samples for AFAM only required a few simple processing steps such as placing a droplet of bacterial suspension on a silicon (Si) wafer and waiting for the samples to dry naturally, whereas, for SEM, the sample processing steps include cleaning, chemical fixation, drying and depositing noble metal onto the cell surface.

Comprehensive, accurate and thorough analysis of acoustic images is also a challenge. Usually, acoustic images in AFAM are separated from the morphological image, and it is difficult to identify signals which are different from surface structures. The morphology images acquired by AFM can highlight cytoplasmic regions, and the acoustic phase and amplitude images may suggest internal structures, similar to the optical fluorescence system with nuclear and cytoplasmic regions separated. Notably, combined AFAM images with RGB-colors presentation could demonstrate detailed, accurate topological information of the physical features, and reduce the image artifacts in the morphology in a more visual and colored manner. For the artifacts caused by diversity of samples’ height in the morphological images, the cell boundary could be clearly outlined by the acoustic phase and amplitude images, while each single color represented one type of image. The multicolor image highlighted subcellular edges by superimposed color from artifacts and noise, while separated color represented extraneous information which showed outside the protruding border. With the fusion method, the atlas images showed distinct morphology and suggested internal structure information in a color perspective.

Ultrasound is rarely used to image nanoscale structures and is more commonly used to image organs deep within the body or within tissues, with the aid of an ultrasound contrast agent. Although SNFUH was developed to detect inner structures at nanoscale resolution in 2005, no prototype that can be widely used in scientific research has been developed, due to difficult maneuverability, e.g., the hard formation of the standing wave [15,16,17,18,19]. Different from the resonance mode of SNFUH, transmission mode is easy to operate and suitable for large-scale use. In transmission AFAM, acoustic images exhibited enhanced edges of morphological information and fewer artifacts of the morphology, highlighted the faint structures in the morphology and suggested different internal areas that will be explored in detail in the future. As a commercialized instrument, AFAM is mature and stable in operating transmission mode. AFAM has been used to image chemical molecular structures, and our group began the research applying transmission AFAM to image cells. As cell imaging using transmission AFAM achieved notable results, this novel approach could be advanced as a general method to experimentally obtaining insight into subcellular mechanisms in a nondestructive and facile manner at a nanoscale resolution.

Recently, the Human Cell Atlas attracts great interests to advance better understanding of human evolution, and help diagnose human diseases by comparing cell types [24,25,26,27]. With extensive imaging range, easy sample preparation, multi-color visualization and nondestructive imaging, the AFAM could advance cell imaging, which would improve the Human Cell Atlas project. Hence, the future applications of AFAM on biological research are promising.

## 5. Conclusions

We introduced a novel imaging method and image fusion algorithm to successfully characterize substructures of various cells, including eukaryotes and prokaryotes, at submicron scale resolution in a nondestructive and convenient manner. The optimized parameters extensively improved the image qualities, and the multicolor-presentation enabled demonstration of both the surface and internal morphology in a single image. Using this system, AFAM was applied to acquire images of samples with a wide range of sizes, whereas optical microscopy and SEM both have imaging range limits. Compared to multiple imaging instruments, AFAM could accurately demonstrate the morphology and certain internal structures of cells in a facile and non-invasive manner. This cell imaging technique will have great practical significance and application prospects in exploring pathogenesis, clinical manifestations and gene lesions due to its high resolution, rapid imaging speed, nondestructive detection nature (transmission), and easy operation. This approach could enhance the potential applications of AFAM in mapping subcellular structures at the nanoscale.

## Figures and Tables

**Figure 1 cells-08-00314-f001:**
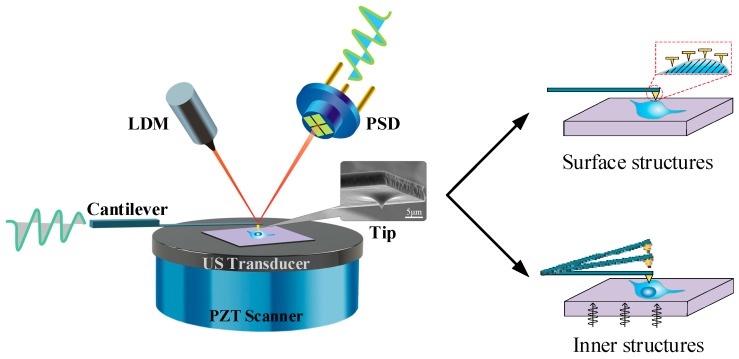
Schematic diagram of transmission atomic force acoustic microscopy (AFAM).

**Figure 2 cells-08-00314-f002:**
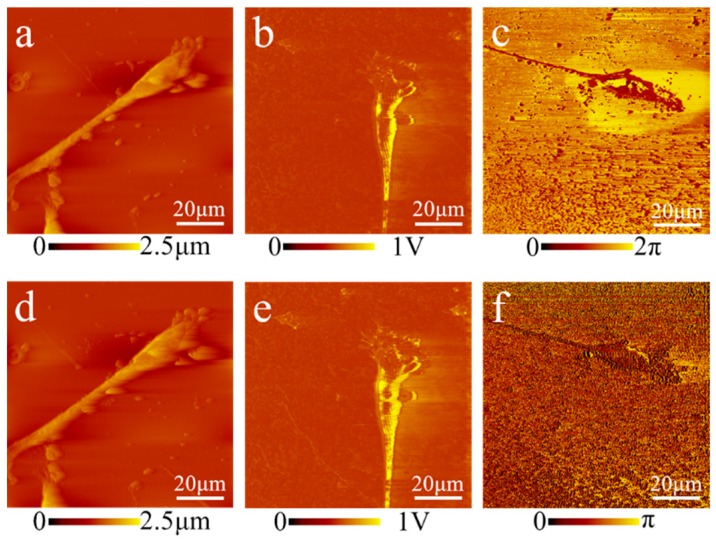
Optimization of parameters for AFAM imaging of MDA-MB-231 cells: (**a**) morphological image obtained at a 0.5 Hz scanning frequency; (**b**) the ultrasonic amplitude image obtained with an integral gain of 300 times, a proportional gain of 200 times and a reference point value of 0.18 V; (**c**) ultrasonic phase images obtained with a 30.5 KHz acoustic frequency; (**d**) the morphological images obtained at a 2 Hz scanning frequency showed blurred edges and more blurred artifacts, compared to image of (**a**); (**e**) the amplitude image obtained with an integral gain of 200 times, a proportional gain of 300 times and a reference point value of 0.14 V showed more artifacts and less clear edges, compared to image of (**b**); and (**f**) the phase image obtained at 1.5 MHz showed less information and more noise than the phase image obtained at 30.5 KHz, compared to image of (**c**). The parameter optimization clearly improved the image quality.

**Figure 3 cells-08-00314-f003:**
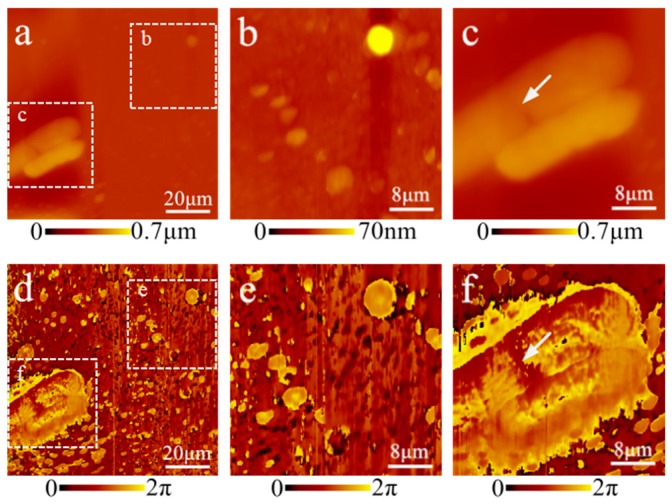
Correlation of AFAM phase images with morphology: (**a**) morphological image of *E. coli*; (**d**) phase image; (**b**) morphological image with 100 times enhancement of the contrast; (**e**) corresponding domain in the phase image with enhanced effects to show the mild structures, including a very bright spot and several dots arranged along a nearly straight line across the figure; (**c**) morphological image of the *E. coli* body structure appeared as a small thin region in the middle’ and (**f**) corresponding domain in the phase image showing more details, including a larger distinct fuzzy area, as indicated by the arrow. The phase image emphasized faint details and indicated subcellular structures inside cells.

**Figure 4 cells-08-00314-f004:**
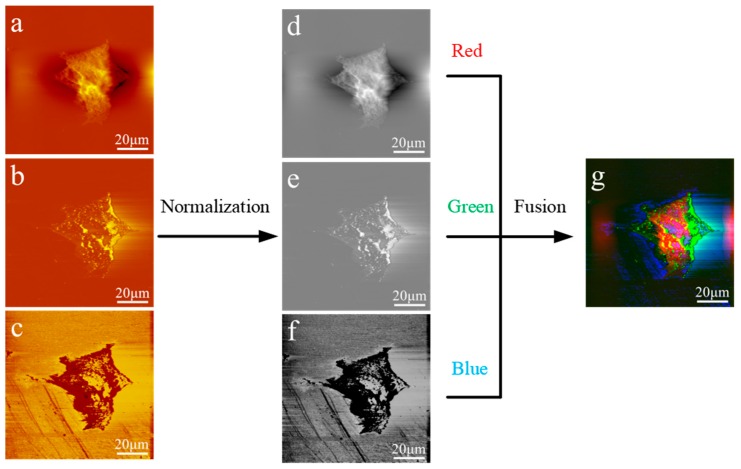
The Fusion method schematic diagram: (**a**) morphological image of an MCF7 cell; (**b**) amplitude image; (**c**) phase image; (**d**–**f**) the source images normalized to create a histogram at the 0–255 level; and (**g**) fused image. The morphological image was taken as the red channel, the ultrasound amplitude image as the green channel, and the ultrasound phase image as the blue channel of RGB. The fused image showed all the information of the original images in a comprehensive and colorful presentation.

**Figure 5 cells-08-00314-f005:**
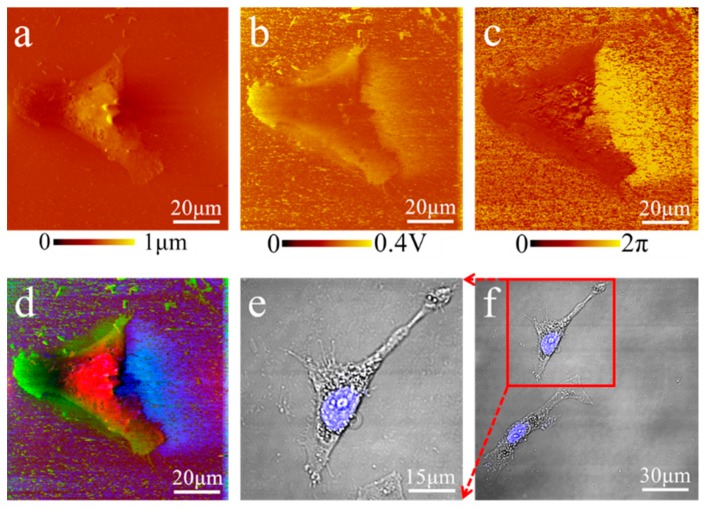
Fused image showing distinct structures: (**a**–**c**) morphological image, amplitude image and phase image of an MDA-MB-231 cell, respectively; (**d**) the fused image; and (**e**,**f**) fluorescence images. The blue region reflected the nucleus with DAPI staining. The fused RGB image had a similar nucleus-like region as in the fluorescence image.

**Figure 6 cells-08-00314-f006:**
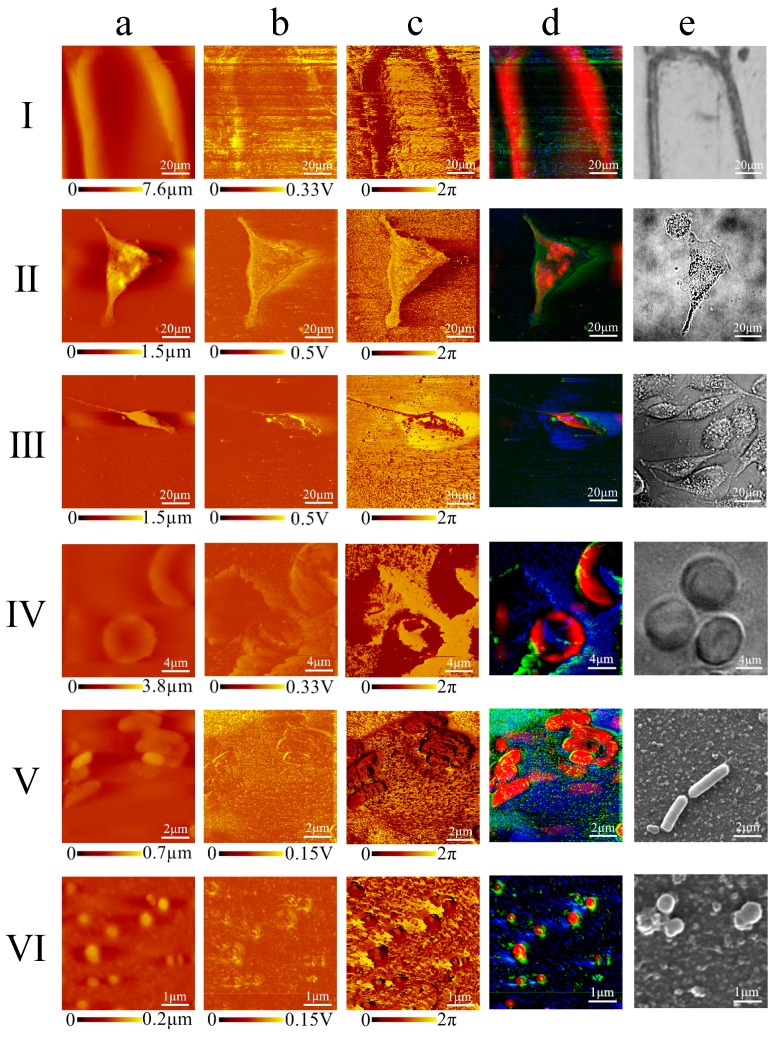
The image atlas of eukaryotes and prokaryotes, presented as a sequence of the morphological image, amplitude acoustic image, phase image, fused image and reference image: (**Ia**–**Ic**) onion epidermal cells have a rectangular-like shape with a cell wall with relatively smooth edges; (**Id**–**Ie**) the fused image emphasizes the cell wall compared to the optical picture; (**IIa**–**IIc**) an MCF7 cell showing localized parapodium and a changeable cell membrane; (**IId**–**IIe**) fused image suggesting some internal structure compared to the optical image; (**IIIa**–**IIIc**) an MDA-MB-231 cell in a classic fusiform shape; (**IIId**,**IIIe**) fused image demonstrating a complicated clear contour compared to the optical image, and suggesting internal information; (**IVa**–**IVc**) human erythrocyte with a cake-like shape that is depressed in the middle; (**IVd**,**IVe**) fused image showing the classic shape, with better resolution than the optical image; (**Va**–**Vc**) *E. coli* showing a representative rod shape; (**Vd**,**Ve**) fused image showing a tridimensional perspective and an image acquired by SEM; (**VIa**–**VIc**) *S. aureus* appeared grape shaped; and (**VId**,**VIe**) fused image showing clearer picture, as good as images acquired by SEM. Transmission AFAM had an imaging scale of 1–100 μm.

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
