# Peer review of "Noninvasive Subcellular Imaging Using Atomic Force Acoustic Microscopy (AFAM)"

_cells, 2019, doi:10.3390/cells8040314_

Round 1
Reviewer 1 Report
The authors describe a method for the imaging of biological samples using acoustic macroscopy. They apply this technique to obtain non—damaged images of biological samples.
The paper is interesting and the conclusions are supported by the experimental data, so I would recommend its publication on “Cells” provided that some MINOR POINTS, including some language issues, highlighted here below are addressed.
Abstract
system ,to à system, to
Introduction
The authors write that “ has poor spatial resolution but strong damage to samples“. They should quantify “poor”. Besides this, I’d write “poor spatial resolutions despite strong damage”, since typically the higher the resolution is, the more damage is created.
Imagesas à images as
Thosetechniques à those techniques
At line 97, “with a 4% solution of paraformaldehyde for 15 minutes to simplify the imaging process”.
Why does paraformaldehyde simplify the imaging process? Can they authors comment on possible artefacts created by this procedure?
Line 154: The à the
Author Response
Manuscript ID: cells-468021
Type of manuscript: Article
Title: Noninvasive subcellular imaging using atomic force acoustic microscopy (AFAM)
Authors: Xiaoqing Li, Ang Lu, Wenjie Deng, Li Su, Jing Wang *, Mingyue Ding *
E-mails: grace@hust.edu.cn, 348411351@qq.com, 782370615@qq.com,
lisu@hust.edu.cn, wang.jing@hust.edu.cn, myding@hust.edu.cn
Dear editor:
We are grateful to the editors and reviewers for their time and constructive comments on our manuscript. We thank both reviewers for their valuable reviews. We have implemented the revisions of their comments and suggestions in the revised version of the manuscript. We provide a point-by-point response to each of the reviewers’ comments below. The changes are also highlighted in yellow in the revised manuscript.
Best regards
Mingyue Ding
E-Mail: myding@hust.edu.cn
Response to Referee 1's Comments
The authors describe a method for the imaging of biological samples using acoustic microscopy. They apply this technique to obtain none—damaged images of biological samples. The paper is interesting and the conclusions are supported by the experimental data, so I would recommend its publication on “Cells” provided that some MINOR POINTS, including some language issues, highlighted here below are addressed.
We thank the referee for the positive appreciations of our work. We addressed below the remaining comments.
Comment 1: The authors write that “has poor spatial resolution but strong damage to samples“. They should quantify “poor”. Besides this, I’d write “poor spatial resolutions despite strong damage”, since typically the higher the resolution is, the more damage is created.
Response to comment 1: We agree with the referee’s comment that the statement is vague and misleading. Now, we rewrote this part in our revised manuscript and add one more reference in our revised manuscript on Page 1 Lines 28-30, to clarify it as follows:
The conventional optical imaging methods have advanced to achieve cell images at nano-resolution, but limited to some complexities, such as the fluorophore localization and fitting algorithms to correct for sample drift during acquisition. [1]
1. Sanderson MJ, Smith I, Parker I, Bootman MD. Fluorescence microscopy. Cold Spring Harb Protoc. 2014, 2014, pdb.top071795. doi: 10.1101/pdb.top071795.
Comment 2: At line 97, “with a 4% solution of paraformaldehyde for 15 minutes to simplify the imaging process”. Why does paraformaldehyde simplify the imaging process? Can they authors comment on possible artefacts created by this procedure?
Response to comment 2: We thank the Referee for pointing out our omission. Here, the cell fixation using the paraformaldehyde is the sample preprocessing before the imaging process, which is just to follow the common cell preparation procedure and to keep the cell shape so that it can be easily compared with other imaging modality. The cell fixation can maintain the cell morphology, make the cells adhere to the wafers more firmly to prevent them from being taken away by the probe during the imaging scan and further make make the scanning process less affected to the sample shift. To more clearly clarify it, we modified the statement on Page 3 Lines 102-106, of the manuscript as follows:
Before AFAM measurement, the cells on the Si wafer was treated with a 4% solution of paraformaldehyde for 15 minutes to fix the cells, which can maintain the cell morphology, make the cells adhere to the wafers more firmly to prevent them from being taken away by the probe during the imaging scan and further make the scanning process less affected to the sample shift.
The treatment with 4% solution of paraformaldehyde is aimed to fix the cells and keep their morphology during the imaging scanning process. This procedure has been commonly used in various optical imaging approaches, and there is no significant artefacts created in the procedure, which has been reported in the reference listed below. We added this description on Page 3 Lines 106-108 of the manuscript, along with the reference.
The fixation treatment with 4% solution of paraformaldehyde has been widely used in other optical imaging approaches, like the fluorescence imaging. There is no significant artefacts created in the procedure, which has been reported.[21]
21. Andrey N. Kuzmin, Artem Pliss, and Paras N. Prasad. Changes in Biomolecular Profile in a Single Nucleolus during Cell Fixation. Anal Chem. 2014, 86, 10909-16. doi: 10.1021/ac503172b.

Reviewer 2 Report
This manuscript describes an interesting new approach to imaging cells. As such, it is worthy of publication in the journal ‘Cells,’ provided the authors address the following criticisms.
1. The acronym SPAM is not unique in the imaging world (and is most famously used and trademarked in the world of cheap processed meats!). Top hits in a google search for ‘SPAM and imaging’ are for Scanning Probe Acceleration Microscopy Chaibva et al. ‘Surface Science Tools for Nanomaterial Characterization’ (2015). Perhaps another acronym for the technique reported in this manuscript, to prevent confusion?
2.The authors should tone down the hyperbole in the manuscript. Moreover, it presents the pros and cons of other techniques in a more neutral fashion. For example, in the Introduction (line 28) the authors state that fluorescence microscopy has a poor spatial resolution, which is not the case. ‘Super-resolution’ fluorescence images and localizes molecules with exquisite, nanometer spatial resolution. Confocal microscopy is not strongly damaging when imaging at the spatial resolutions seen in SPAM images. In the same paragraph; the authors do not discuss other competing techniques, such as Transmission Electron microscopy/tomography or soft x-ray tomography. Both of which are capable of imaging cells at high spatial resolution. This omission should be corrected.
3.In the opening sentence of the abstract, the authors claim their new technique is ‘noninvasive’ and ‘non-destructive,’ yet they imaged chemically fixed cells! Why? Ideally, I would like to see this work repeated with un-fixed, native state cells before publication.
4.Figure 6 I a-e. The reference image for the onion epidermal cells (Fgi. 6I e) has an order of magnitude larger field of view than the other images in the series. This makes a comparison between the SPAM images and the reference impossible. The authors should either include a similar field-of-view reference image or remove this work from the manuscript completely.
5.Materials and Methods (line 106 -112). The authors should give more detailed information on the bacterial cell growth – were the cells in log phase or stationary phase? Why are the bacteria referred to as ‘bacterial complexes’ (line 110)? Moreover, why were the bacteria mounted on Si wavers exposed to an open flame for 200 ms? Was this step checked for damage to the cells using a light microscope?
6. The grammar and language could be improved. For example, the opening sentence in the abstract could be shortened and less repetitive ("noninvasive biological imaging approaches.." and "nondestructive imaging of cells" are, more or less, the same thing). On line 319, there is a typo -- 'placing' or similar should replace 'smashing' when describing the application of the bacterial suspension on the Si wafer.
Author Response
Manuscript ID: cells-468021
Type of manuscript: Article
Title: Noninvasive subcellular imaging using atomic force acoustic microscopy (AFAM)
Authors: Xiaoqing Li, Ang Lu, Wenjie Deng, Li Su, Jing Wang *, Mingyue Ding *
E-mails: grace@hust.edu.cn, 348411351@qq.com, 782370615@qq.com,
lisu@hust.edu.cn, wang.jing@hust.edu.cn, myding@hust.edu.cn
Dear editor:
We are grateful to the editors and reviewers for their time and constructive comments on our manuscript. We thank both reviewers for their valuable reviews. We have implemented the revisions of their comments and suggestions in the revised version of the manuscript. We provide a point-by-point response to each of the reviewers’ comments below. The changes are also highlighted in yellow in the revised manuscript.
Best regards
Mingyue Ding
E-Mail: myding@hust.edu.cn
Response to Referee #2's Comments
This manuscript describes an interesting new approach to imaging cells. As such, it is worthy of publication in the journal ‘Cells’, provided the authors address the following criticisms.
We thank the referee for his/her positive appreciations of our work and provide us an opportunity to polish it. We addressed the remaining concerns below.
Comment 1: The acronym SPAM is not unique in the imaging world (and is most famously used and trademarked in the world of cheap processed meats!). Top hits in a google search for ‘SPAM and imaging’ are for Scanning Probe Acceleration Microscopy Chaibva et al. ‘Surface Science Tools for Nanomaterial Characterization’ (2015). Perhaps another acronym for the technique reported in this manuscript, to prevent confusion?
Response to comment 1: We agree with the referee and now replace the acronym by AFAM. We did a research of the acronyms of similar Imaging approaches, found that AFAM is used more popularly, and listed some of the references are below.
1. Flores-Ruiz FJ, Espinoza-Beltrán FJ, Diliegros-Godines CJ, Siqueiros JM, Herrera-Gómez A. Atomic force acoustic microscopy: Influence of the lateral contact stiffness on the elastic measurements. Ultrasonics. 2016, 71:271-277. doi: 10.1016/j.ultras.2016.07.003.
2. Kimura K, Kobayashi K, Yao A, Yamada H. Visualization of subsurface nanoparticles in a polymer matrix using resonance tracking atomic force acoustic microscopy and contact resonance spectroscopy. Nanotechnology. 2016,27, 415707. doi: 10.1088/0957-4484/27/41/415707.
3. Kopycinska-Müller M, Clausner A, Yeap KB, Köhler B, Kuzeyeva N, Mahajan S, Savage T, Zschech E, Wolter KJ. Mechanical characterization of porous nano-thin films by use of atomic force acoustic microscopy. Ultramicroscopy. 2016, 162, 82-90. doi: 10.1016/j.ultramic.2015.12.001.
4. Y. Luo, M. Büchsenschütz-Göbele, W. Arnold, K. Samwer. Local elasticity and mobility of twin boundaries in martensitic films studied by atomic force acoustic microscopy. New J. Phys. 2014, 16, p. 013034, 10.1088/1367-2630/16/1/013034
Comment 2: The authors should tone down the hyperbole in the manuscript. Moreover, it presents the pros and cons of other techniques in a more neutral fashion. For example, in the Introduction (line 28) the authors state that fluorescence microscopy has a poor spatial resolution, which is not the case. ‘Super-resolution’ fluorescence images and localizes molecules with exquisite, nanometer spatial resolution. Confocal microscopy is not strongly damaging when imaging at the spatial resolutions seen in SPAM images. In the same paragraph; the authors do not discuss other competing techniques, such as Transmission Electron microscopy/tomography or soft x-ray tomography. Both of which are capable of imaging cells at high spatial resolution. This omission should be corrected.
Response to comment 2: We accept the criticisms from the referee. We made several modifications in the manuscripts to fix them. We have rewritten the description about the fluorescence microscopy on Page 1 Lines 28-30 and add a reference in our revised manuscript as follows:
The conventional optical imaging methods have advanced to achieve cell images at nano-resolution, but limited to some complexities, such as the fluorophore localization and fitting algorithms to correct for sample drift during acquisition. [1]
1. Sanderson MJ, Smith I, Parker I, Bootman MD. Fluorescence microscopy. Cold Spring Harb Protoc. 2014, db.top071795. doi: 10.1101/pdb.top071795.
Moreover, other competing techniques, such as Transmission Electron microscopy/tomography or soft x-ray tomography can image cells at nano-resolution. We added it in revised manuscript on Pages 1-2, Lines 39-46 as well as the references.
The transmission electron microscopy can be used to visualize some virus particles and cells at nano-resolution. [14] The soft X-ray tomography can also be used for the quantitative imaging of organelle structure and distribution in whole cells. [15] Though those techniques have achieved nanometer resolution on cell imaging, they both have their own limitations. For instance, AFM images have insoluble artifacts caused by the abrupt height of the samples and can only obtain the surface or sub-surface structure, [7-12] SEM and TEM has special requirements for sample preparation, [13-14] and the soft X-ray tomography has potential radiation safety limitation. [15]
14. Noda T. Electron Microscopy of Ebola Virus-Infected Cells. Methods Mol Biol. 2017, 1628, 243-250. doi: 10.1007/978-1-4939-7116-9_19.
15. Parkinson DY, McDermott G, Etkin LD, Le Gros MA, Larabell CA. Quantitative 3-D imaging of eukaryotic cells using soft X-ray tomography. J. Struct. Biol. 2008, 162, 380-6. doi: 10.1016/j.jsb.2008.02.003.
Comment 3: In the opening sentence of the abstract, the authors claim their new technique is ‘noninvasive’ and ‘non-destructive,’ yet they imaged chemically fixed cells! Why? Ideally, I would like to see this work repeated with un-fixed, native state cells before publication.
Response to comment 3: Here we claim it non-invasive and non-destructive, because both the scanning process and the acoustic wave will not bring damages to the samples. Moreover, this cell fixation is not inevitable. We fixed the cells is just to follow the common cell preparation procedure to keep the cell shape, in order to compare with other imaging modality more easily. The treatment with 4% solution of paraformaldehyde is aimed to fix the cells and keep their morphology during the imaging scan, which is just a preprocessing of cells cultured in medium. Furthermore, the onion epidermis cells and erythrocytes in Figure 6 were un-fixed and in native state, which were directly observed by AFAM. We revised the description on Page 3 Lines 124-129 of the manuscript below.
We chose the six samples mentioned above to test the imaging capacity of our method, in which MDA-MB-231, MCF7 cells, E. coli and S.aureus were fixed, while the onion epidermis cells and erythrocytes were unfixed and in native. The treatment with 4% solution of paraformaldehyde or the flame preprocessing is aimed to fix the cells and keep their morphology during the imaging scan, which is just a preprocessing of cells cultured in medium.
Comment 4: Figure 6 I a-e. The reference image for the onion epidermal cells (Fig. 6I e) has an order of magnitude larger field of view than the other images in the series. This makes a comparison between the SPAM images and the reference impossible. The authors should either include a similar field-of-view reference image or remove this work from the manuscript completely.
Response to comment 4: We have accepted the suggestion of the referee and replaced the reference image of the onion epidermal cells in the same field-of-view of AFAM images for better comparison as shown in Figure 6, on Page 10 of our revised manuscript.
Comment 5: Materials and Methods (line 106 -112). The authors should give more detailed information on the bacterial cell growth – were the cells in log phase or stationary phase? Why are the bacteria referred to as ‘bacterial complexes’ (line 110)? Moreover, why were the bacteria mounted on Si wavers exposed to an open flame for 200 ms? Was this step checked for damage to the cells using a light microscope?
Response to comment 5: After cultured in Luria-Bertani (LB) media in an incubator shaker at 37 °C and 250 rpm for 12 hours, the bacterial cells were in the log phase. We added this description into Lines 118-121 on Page 3 of the manuscript.
E. coli and S. aureus, stored by the China Center for Type Culture Collection in Wuhan (Wuhan H22), were taken out of the -20 °C refrigerator. After cultured in Luria-Bertani (LB) media in an incubator shaker at 37 °C and 250 rpm for 12 hours, the bacterial cells were in the log phase.
The ‘bacterial complexes’ referred to the culture medium with the bacteria, which has been rewritten in the manuscript (Lines 121-122, Page 3).
A total of 50 μL medium of the bacteria was added to a 1.5 mL EP tube and centrifuged for 4 minutes at 10000 rpm/min and 4 °C.
The bacteria mounted on Si wavers exposed to an open flame for 200 ms is to make the bacteria adhere to the wafers more firmly to prevent them from being taken away by the probe as it scans. After the flame treatment, the bacteria are dead and fixed on the Si wafer. We had checked the prepared samples under a regular confocal microscope, which is unable to show the difference due to its limited resolution. Moreover, the samples of the SEM reference image of bacteria were processed by the same procedure, which showed the morphology has no change. Actually, this preprocessing step is not a necessary for sample preparation, which has the same function with the treatment with 4% solution of paraformaldehyde. The fixation is just to follow the common cell preparation and kept the cell shape so that it can be easily compared with other imaging modality. Moreover, another experiment that we mounted the E. coli on the Si wafers without exposure to the flame, was supplemented to check if this sample preprocessing is necessary. The experimental results confirmed that the fixation is not inevitable. (The experimental figures can be seen in the attached Author's Notes File.) We have revised the text on Page 3 Lines 127-132 in our revised manuscript to clarify it further.
The treatment with 4% solution of paraformaldehyde or the flame preprocessing is aimed to fix the cells and keep their morphology during the imaging scan, which is just a preprocessing of cells cultured in medium. Or in other words, this imaging approach does not require the cell fixation as an necessary process step, which we have checked though a supplemented experiment. We fixed the cells to follow the common cell preparation, which kept the cell shape so that it can be easily compared with other imaging modality.
Comment 6: The grammar and language could be improved. For example, the opening sentence in the abstract could be shortened and less repetitive ("noninvasive biological imaging approaches." and "nondestructive imaging of cells" are, more or less, the same thing). On line 319, there is a typo -- ‘placing’ or similar should replace 'smashing' when describing the application of the bacterial suspension on the Si wafer.
Response to comment 6: Done.
